# Center of Pressure Deviation during Posture Transition in Athletes with Chronic Ankle Instability

**DOI:** 10.3390/ijerph20085506

**Published:** 2023-04-14

**Authors:** Takanori Kikumoto, Shunsuke Suzuki, Tomoya Takabayashi, Masayoshi Kubo

**Affiliations:** 1Institute for Human Movement and Medical Sciences, Niigata University of Health and Welfare, 1398 Shimami-cho, Kita-Ku, Niigata City 950-3198, Niigata, Japan; 2Department of Physical Therapy, Niigata University of Health and Welfare, 1398 Shimami-cho, Kita-Ku, Niigata City 950-3198, Niigata, Japan

**Keywords:** chronic ankle instability, center of pressure deviation, posture transition, ankle strategy, joint dysfunction, knee joint immobilization

## Abstract

Center of pressure (COP) tracking during posture transition is an ideal scale for determining the recurrence of an ankle injury, thereby preventing chronic ankle instability (CAI). However, the same is difficult to determine because the reduced ability of certain patients (who experienced sprain) to control posture at the ankle joint is masked by the chain of hip and ankle joint motion. Thus, we observed the effects of knee joint immobilization/non-immobilization on postural control strategies during the posture transition task and attempted to evaluate the detailed pathophysiology of CAI. Ten athletes with unilateral CAI were selected. To examine differences in COP trajectories in the CAI side and non-CAI legs, patients stood on both legs for 10 s and one leg for 20 s with/without knee braces. COP acceleration during the transition was significantly higher in the CAI group with a knee brace. The COP transition from the double- to single-leg stance phase was significantly longer in the CAI foot. In the CAI group, the fixation of the knee joint increased COP acceleration during postural deviation. This suggests that there is likely an ankle joint dysfunction in the CAI group that is masked by the hip strategy.

## 1. Introduction

Lateral ankle sprain (LAS) is a prevalent musculoskeletal injury that occurs frequently during sports activities [1]. This type of injury can result in substantial medical expenses [2] and long-term consequences [3], with approximately 40% of individuals with LAS developing chronic ankle instability (CAI) [4]. CAI is defined as a giving way or subjective instability in the ankle joint and a history of recurrent sprains [5]. CAI causes functional impairments, such as reduced ankle extensor strength, reduced dynamic balance ability, and delayed peroneus longus reaction time [6], affecting performance. Among the functional-phase disorders listed above, a decrease in postural control after LAS leads to a recurrence of LAS and is considered as a pathway to CAI. It is necessary to use an assessment of postural control ability to identify individuals at high risk for recurrent ankle sprain prior to the onset of CAI and to initiate appropriate rehabilitation.

Postural control ability is defined as the ability to maintain, achieve, or regain a state of balance in any upright position [7]. Previous studies have suggested that changes in stability of standing postural control may be a clue in identifying individuals at high risk for LAS recurrence such as CAI. For example, it has been reported that individuals at high risk for ankle sprains have a higher center of pressure (COP) displacement during standing tasks [8], and recurrent ankle sprains cause CAI. In addition, a decrease in postural control ability has been reported in the CAI group compared to the healthy group in a single-leg standing task [9]. However, it is difficult to selectively capture the contribution of ankle instability to single-leg standing postural control from changes in COP variables alone. One reason for this may be that COP variables change in single-leg standing postural control as a result of multiple joint strategies.

Therefore, it has been suggested that changes in COP need to be evaluated along with changes in movement strategies between body segments [10]. We hold the view that the double-leg stance to single-leg stance transition (DLS-SLS test), which involves the transition from a double-leg stance to a single-leg stance, represents a prime example of the connection between movement and postural stability, and it can serve as a discriminative measure to assess CAI impairment. In addition, it has been reported that healthy young adults predominantly use the ankle joint strategy to maintain balance [9,11], while the CAI group predominantly uses the hip joint strategy [9]. In light of the above, there are many unknowns in selectively assessing ankle joint strategies in the DLS-SLS test, which requires an interaction between locomotion and postural stability. In order to address the aforementioned issues, we deemed it pertinent to investigate the degree of neuromuscular control noise using Fitts’ model, as we subscribe to the notion that it is the fundamental factor that shapes motor characteristics based on the principle of speed (transition) versus accuracy trade-off. Fitts’ model is a model of human movement, which makes it possible to predict that the time required to move to the target area is a function of the distance to the target part and the size of the object.

The purpose of this research was to examine the impact of various positions on the assessment of postural control in individuals with ankle instability. It was suggested that in the SLS position with knee flexion, the loss of postural control ability at the ankle joint was difficult to observe as it was masked by the chain of hip and ankle joint motion. To overcome this limitation, the study evaluated postural control ability in the SLS position with the knee fixed in extension. The hypothesis was that hidden ankle instability would be captured more prominently in this position, which could serve as a screening indicator for identifying individuals at high risk of re-injury. By comparing the results from both positions, the study aimed to provide valuable insights into developing effective rehabilitation programs for ankle instability.

Furthermore, several prior studies have utilized distinct subject groups for those with CAI and those with healthy ankles. However, postural control ability can vary significantly between individuals and is contingent upon factors, such as ankle joint range of motion and proprioceptive sensation [12]. It is also highly dependent on the history of injury and severity, as well as the mental fatigue level at the time of the examination [13]. Therefore, this study focused on athletes who developed CAI in only one leg and examined the left–right difference.

The purpose of this research was to investigate how the function of the knee joint affects the DLS-SLS test on the opposite side to the CAI in individuals who suffer from CAI in only one foot. The primary goal was to detect any ankle instability that might be hidden by the movement of the hip joint chain. Our hypothesis was that the CAI group, with their knee joint fixed in extension, would exhibit a more significant deviation in the center of pressure than the non-CAI group. This difference would indicate a concealed phenomenon that would remain undetected otherwise. By gaining a deeper understanding of the pathophysiology of CAI, we can develop more effective evaluation methods to identify individuals who are at a higher risk of re-injury from ankle sprains.

## 2. Participants and Methods

### 2.1. Participants

From Niigata University of Health and Welfare, we surveyed 186 male students using the International Ankle Consortium guidelines to assess subjective ankle joint instability. Out of the total number of participants, 23 individuals (12.43%) were included in the CAI group based on the inclusion criteria. Moreover, 10 participants (43.2%) from the symptomatic group were found to have CAI in only one foot, based on the same criteria. The inclusion criteria were: (1) a score of 25 or less on the Cumberland Ankle Instability Tool (CAIT) (Japanese version) [14], a recommended questionnaire for subjective instability; (2) a history of at least one LAS; (3) a history of giving way, instability, or sprain; (4) no episode of ankle sprain within 3 months from the date of measurement. Subjects who met the above four criteria were selected. Ten subjects with CAI in only one foot had a Tegner Activity Scale score of 9 or higher. The contralateral leg selected for the CAI group was defined as the non-CAI side that was not included in the CAI inclusion criteria and had not sprained the ankle for more than one year. Participants who had undergone surgery on their lower extremities in the past and those who expressed clear signs of fatigue were not included. The study was conducted in accordance with the Declaration of Helsinki after being approved by the ethics committee at Niigata University of Health and Welfare (approval No. 18583-210218). The study information was thoroughly presented to the subjects, and all subjects submitted written informed consent before participating in the study.

### 2.2. Procedure

The experimental task consists of the DLS-SLS test, which entails a 10 s barefoot double-leg standing phase followed by a 20 s single-leg standing phase, wherein the participant is required to maintain their heel alignment with respect to a reference line and keep their arms relaxed at their sides. During the practice test, a goniometer was used to measure the elevating leg to unify the flexion of 30 degrees at the hip joint and 45–50 degrees at the knee joint during the holding of the single-leg stand to improve reproducibility. The knee joint of the supporting leg was subjected to the following two conditions: (1) in the immobilized condition, the subject’s knee joint was immobilized with a knee brace (SecuTec Genu, Bauerfeind AG, Zeulenroda-Triebes, Germany) and held at 0 degrees of knee joint extension, and (2) the unfixed condition was performed under conditions that allowed the subject to bend the knee normally. More information regarding the experimental layout is provided in Figure 1, which outlines the specifics of the research design. The trial task was performed randomly in the knee joint fixation and non-fixation conditions. Three sets of three trials were performed for each condition (Figure 2), alternating between the CAI and non-CAI support conditions. To exclude the influence of fatigue, a one-minute rest was taken between each trial and a three-minute rest between sets. The subjects practiced for 30 s before the measurement as a practice. The kinematic features pertaining to the COP deviation, such as the range and mean of displacement, as well as the maximum velocities, were employed to characterize the DLS-SLS transition.

If any of the following occurred during the measurement—(1) the raised lower limb touched the floor, (2) the arms were unclasped, (3) holding time of single-leg standing position was less than 20 s—the trials were considered failed trials. COP deviations during the trial were measured at 300 Hz using a plantar pressure distribution measurement system (FootScan Entry Level Systems: Rsscan) that guaranteed accuracy and reliability [15].

### 2.3. Data Analysis

In this study, we tried to verify the COP deviation and examine the area and movement acceleration. To analyze the DLS-SLS transition, we used Fitts’ model, which predicts that the time required to move to the region of a target that models human movement is a function of the distance to the target and its size. According to the assumptions underpinning Fitts’ model, achieving postural stability during the DLS-SLS transition necessitates moving the COP as expeditiously as feasible from its initial position to a target point located at a distance D with a width equal to that of the participant’s foot (*W*) [16]. It is predicted that the duration necessary to expeditiously displace the COP from the DLS to the SLS target position would be influenced by the ratio between the distance separating the initial position and the target point (*D*) and the width of the target area (*W*). The logarithm of this ratio is used to derive the travel difficulty index (*ID*) of the DLS-SLS transition. Additionally, we hypothesized that the increased amplitude of COP displacement at the target position would be regarded as a form of noise (*N*) in neuromuscular control, resulting in a higher *ID* value [17].
(1)$$ID=\log_{2}\left(\frac{2D}{W-N}\right)$$

### 2.4. Statistics

All statistical analyses were performed using R studio software. The difference between the subjective instability score and the time of the last ankle sprain was examined using a paired T-test. Regarding COP, the normality of distribution for quantitative data was assessed using the Kolmogorov–Smirnov test. In order to identify whether there were statistically significant differences between the feet (CAI vs. non-CAI) and the two conditions (baseline vs. knee brace), a repeated measures analysis of variance (ANOVA) was employed. A *p*-value of <0.05 was considered significant.

## 3. Results

### 3.1. Correlation Analysis

This study was conducted on male athletes who exercised regularly, with an average age of 20.5 ± 0.5 years. The mean body height of the participants was 180.1 + 1.5 cm, and their body weight was 83.0 ± 1.0 kg. The length and width of the participants’ feet were also measured, and the group means were calculated. The average foot length was 276.1 ± 10.4 mm, while the average width of the foot was 103.25 ± 6.1 mm. Out of the ten participants, six were diagnosed with certified CAI in their right foot, while the other four had it in their left foot. However, there were no significant differences in foot length and width between the two groups. This information indicates that any observed differences in the DLS-SLS test results were unlikely to be due to differences in foot dimensions.

The CAIT score, a questionnaire to investigate subjective ankle instability, was 20.5± 0.5 for the CAI side and 28.0 ± 1.0 for the non-CAI side, indicating significant subjective instability on the CAI side (*p* < 0.05). The last ankle sprain according to the subjective injury history questionnaire was 3.0 ± 0.5 months on the CAI side and 18.0 ± 2.5 months on the non-CAI side, with a significant difference between the two groups (*p* < 0.01) (Table 1).

### 3.2. Movement Range (D) and Sway Range (N) of Center of Pressure

The ML COP sway range (Figure 3) during the double-leg standing phase remained consistent at the same level of 3.7 ± 4.1 mm and 3.8 ± 5.8 mm for both baseline and knee brace conditions in each trial. Additionally, no significant difference was found between the CAI side (3.7 ± 3.6 mm) and the non-CAI side (3.9 ± 4.3 mm). Similarly, the AP range of sway during the DLS phase was at a level of 4.1 ± 2.8 mm and 4.6 ± 3.3 mm when testing with and without knee brace. No significant difference was found between the CAI side (4.2 ± 2.6 mm) and the non-CAI side (4.4 ± 2.3 mm). There was no significant difference in COP between the double-leg phases, and we do not believe that using the knee brace had any effect on the subsequent single-leg phase.

### 3.3. COP Acceleration during the Transition to SLS 

COP acceleration during the transition to SLS was significantly higher in the CAI group with knee brace than in the non-CAI group and the CAI group without knee brace (*p* < 0.01) (Figure 4). The distance from DLS to SLS for COP transition in the CAI group was significantly higher than that of the non-CAI group, regardless of with knee brace or not (*p* < 0.05) (Figure 5). After the transition to SLS, the M-L distance was significantly higher in the CAI group than in the non-CAI group (*p* < 0.05) (Figure 6), and the A-P distance was significantly higher in the non-CAI group than in the CAI group (*p* < 0.05) (Figure 7), regardless of with knee brace or not.

### 3.4. Movement Index of Difficulty (ID) of Center of Pressure

The difficulty index (ID) in the Fitts’ model can be estimated from the measurement of the player’s foot width (W) and the COP displacement distance (D) during the DLS-SLS transition (Equation (1)). Fitts’ model ID scores were 1.315 for the CAI group and 1.379 for the CAI group with knee braces, and 1.071 for the non-CAI group and 1.113 for the non-CAI group with knee braces. Fitts’ model ID was significantly higher in the CAI group than in the non-CAI group, both with and without the use of knee braces (*p* < 0.05).

## 4. Discussion

The primary goal of the study was to assess how the COP transitions during DLS-SLS differed for individuals with CAI, with and without knee braces. Our aim was to investigate how knee braces impact postural stability control and to determine the role of the knee joint in this task. We compared the performance of participants with and without knee braces to gain insight into the effects of knee braces on the task. This study reduced the involvement of the kinetic chain between the lower extremity joints during the COP transitions from DLS to SLS. To evaluate the dynamic motor control ability more prominently at the ankle joint, a single-leg standing task was performed under two conditions: knee joint immobilization and non-immobilization. The kinetic chain was interrupted using knee braces, and only ankle function in the CAI group was assessed to look for indicators that could identify those most likely to re-injure. Furthermore, we are of the opinion that this verification process will provide us with a comprehensive understanding of the pathophysiology of CAI, and aid in the development of a post-onset CAI rehabilitation program.

Considering the contribution of knee joint motion to standing postural control, the knee joint immobilization in the single-leg standing task used in this study reduces the degrees of freedom for motor control [18]. This is considered to increase the difficulty in the balancing task. The changes are captured by changes in the COP, which contain superimposed information on the motion of multiple joints. The postural control strategy was also changed because of the knee immobilization, which broke the kinetic chain between the lower extremity joints. The increase in acceleration during postural transition observed in the CAI group with knee joint immobilization indicated a shift to a balancing strategy that utilized more hip motion [19]. It has been stated in previous studies that ankle joint strategy plays a major role in the early stages of postural control [20]. In the CAI group, where ankle joint function is thought to be impaired, the medial–lateral control distance was greater during COP deviation, and, therefore, acceleration was also higher. Furthermore, the increase in variables at the feet (A-P) in the non-CAI group suggested that knee joint fixation in the single-leg standing task also affected the maintenance of balance in the anterior–posterior direction. In other words, it is likely that the medial–lateral transfer movements performed in this experiment did not achieve sufficient stability in the CAI group and did not lead to the introduction of an anterior–posterior strategy. This dysfunction would also be expected to increase the load on tissues that contribute to dynamic stability in the medial–lateral direction.

The CAI group significantly used the M-L range of COP during postural control in single-leg standing compared to the non-CAI group. This result may be due to the functional disruption of the ankle joint caused by the onset of CAI; abnormal joint position sense [21] and reduced motor control ability [22] have been observed with the onset of CAI. In addition, there are previous studies describing delayed reaction time of the peroneal muscles in the CAI group [23]. These dysfunctions led to the results of this study. When the knee joint was immobilized in the CAI group, the acceleration of COP during the transition to SLS was significantly higher than when the knee joint was not immobilized. This supports previous findings [9] that more detailed ankle joint dysfunction is revealed by single-leg postural control with knee immobilization, and that the CAI group uses other joint strategies. In addition, the CAI group required a greater contribution from the hip and ankle kinetic chain than the non-CAI group in order to maintain balance in the single-leg stance. This revealed an ankle joint dysfunction masked by the hip strategy, which could not be ascertained by previous pathological examination of CAI. 

The present study examined the effects of knee immobilization on postural control strategies in individuals with CAI during single-leg standing hold. The findings revealed that knee joint immobilization increased the contribution of hip control strategy, as demonstrated by the A-P strategy of COP. Specifically, the CAI group spent more time compensating for the M-L strategy, indicating a decreased reliance on the A-P strategy. The A-P strategy is known to increase the reliance on the hip joint strategy to maintain balance. Additionally, the results showed that knee immobilization unmasked previously undetected ankle joint dysfunctions in the CAI group, which were not apparent in other conditions. These findings are important since impaired balance ability increases the risk of re-injury from ankle sprains, and the CAI group’s heavy reliance on the hip joint strategy during postural control may result in delays during landing and cutting movements, leading to re-injury [24]. Therefore, rehabilitation programs targeting improvements in postural control ability specific to ankle joint strategies alone are needed. This study highlights the importance of understanding the different postural control strategies in individuals with CAI and the impact of knee immobilization on the control strategy. The findings suggest that knee immobilization may have a negative impact on postural control in CAI patients and that it may unmask previously undetected ankle joint dysfunctions. Future research should focus on developing rehabilitation programs that target specific control strategies and evaluating their effectiveness in improving postural control ability and reducing the risk of ankle sprains among CAI patients.

This study had several limitations. Firstly, the small sample size used in this study had a significant impact on the validity of the statistical interpretation of the results. Although individual differences in postural control ability were taken into account, the selection of subjects was limited to athletes with CAI in only one foot. Therefore, it is necessary to conduct a larger-scale study and collect a greater number of subjects in the future. Secondly, this study was a cross-sectional study and did not determine whether the CAI group had a higher re-injury rate than the non-CAI group. Thus, further validation, including prospective studies, may help to identify high-risk groups for ankle sprains based on the above points. Longitudinal studies can help to establish the causal relationship between the variables, whereas cross-sectional studies can only provide an overview of the situation. Thirdly, it is important to note that this study only included male subjects, and women were not included as subjects in this research. This is because the menstrual cycle is known to modulate ankle ligament laxity, and including women would have introduced additional confounding variables. To eliminate the various factors that can modulate laxity, only male subjects were included in this study. However, future studies should also include female subjects to ensure that the findings can be generalized to both genders. Finally, the assessment index did not take into account the subjects’ foot morphology and knee joint alignment, which could have had an impact on the results. Although preliminary validation was conducted because the effects of flatfoot and medial knee could not be eliminated, it was not feasible to incorporate these variables into the study due to difficulties in securing the subjects. Hence, future research should focus on conducting a detailed evaluation of foot morphology and knee joint alignment as determinants to obtain more accurate results. The inclusion of such variables may provide a more comprehensive understanding of the relationship between CAI and postural control ability.

## 5. Conclusions

In this study, we aimed to explore the impact of knee immobilization and non-immobilization with knee brace on the postural control of CAI patients during postural transfer. The results of our study shed light on the changes in balance capacity caused by ankle instability by comparing postural control strategies under knee joint constraint conditions. Our findings revealed that ankle joint dysfunction, which may be masked by the hip joint strategy, was clearly evident in the knee immobilization condition. Moreover, using a knee brace for evaluation may uncover a more detailed pathophysiology of CAI. The CAI group exhibited increased COP acceleration during transfers, which suggests that they may use different strategies than the non-CAI group, putting them at higher risk of re-injuring the ankle. Based on our results, it can be inferred that the reliance of the CAI group on the hip joint strategy for postural control may cause delays in controlling their posture during landing and cutting movements, thereby increasing the risk of re-injury of the ankle sprain. Therefore, rehabilitation interventions aimed at improving balance function are of the utmost importance to enhance postural control ability that is specialized for the ankle joint strategy in CAI patients. This study also emphasizes the need for further research to identify new interventions that can improve postural control in patients with CAI.

## Figures and Tables

**Figure 1 ijerph-20-05506-f001:**
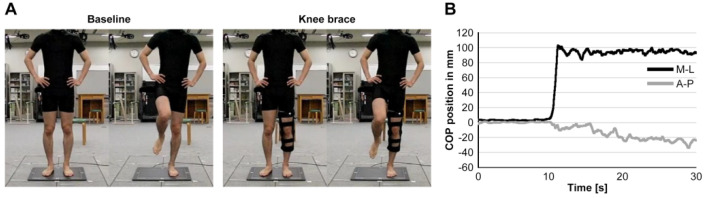
Initial and target posture of the DLS-SLS test ((**A**): Baseline, Knee brace) and the mediolateral (ML) and anteroposterior (AP) COP trajectories (**B**).

**Figure 2 ijerph-20-05506-f002:**
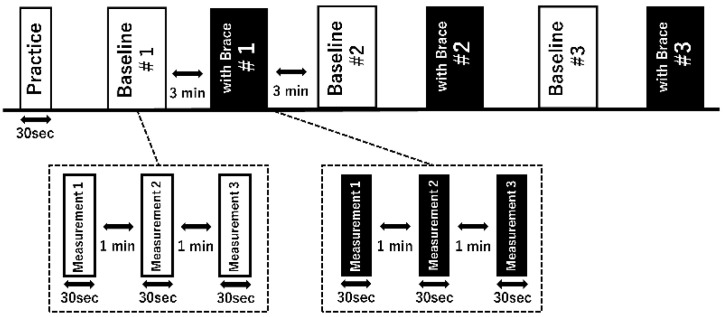
Experimental procedures: one set consisted of three measurements for 30 s and alternately performed with and without the brace, and the first set was randomized with or without brace.

**Figure 3 ijerph-20-05506-f003:**
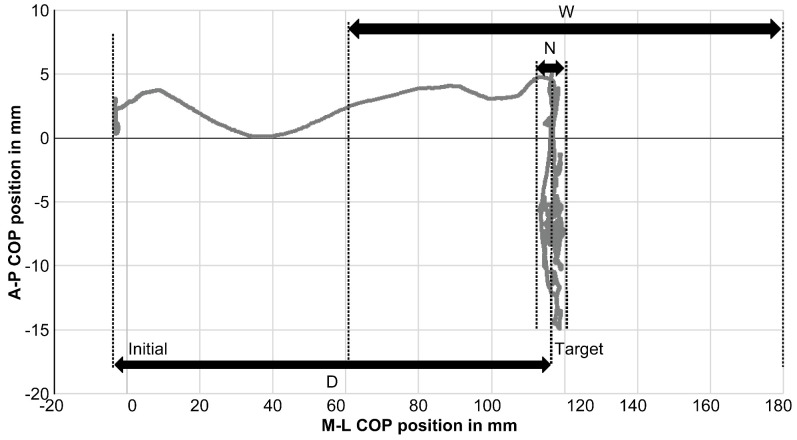
Assessing the degree of center of foot pressure (COP) deviation when transitioning from the initial position (DLS) to a less stable SLS target, using the Fitts’ model depicted in Figure 1 and Equation (1). The index of difficulty is determined by the distance (*D*) between the DLS and SLS equilibrium points, as well as the width of the target relative to the foot width (*W*).

**Figure 4 ijerph-20-05506-f004:**
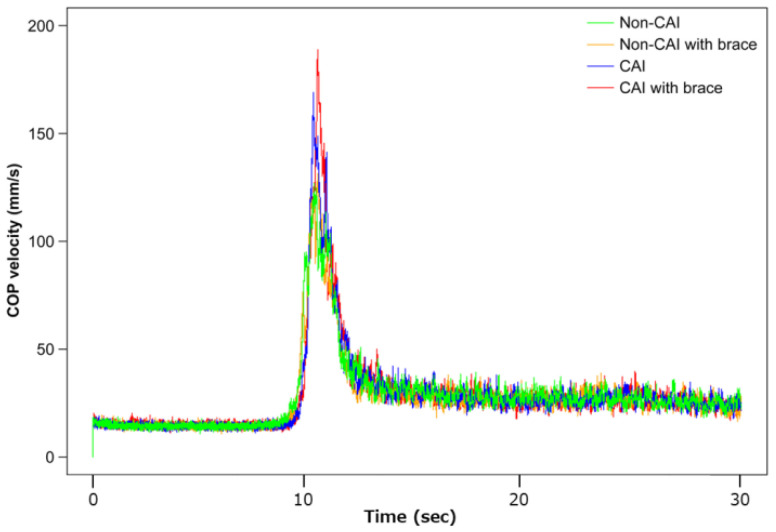
Graph showing the amount of change in center of pressure (COP) velocity from DLS to SLS. The horizontal axis indicates DLS and SLS, and the vertical axis indicates COP velocity. There was a significantly higher score in the CAI group with knee brace than in non-CAI group and the CAI group without knee brace (*p* < 0.01).

**Figure 5 ijerph-20-05506-f005:**
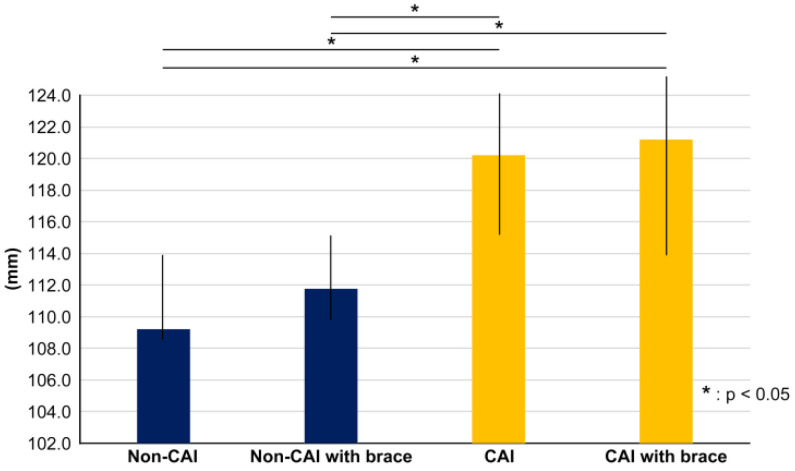
Graph showing the amount of change in transition distance of the center of pressure. There was a significant difference between the CAI and non-CAI sides, but there was no significant difference between with or without the brace.

**Figure 6 ijerph-20-05506-f006:**
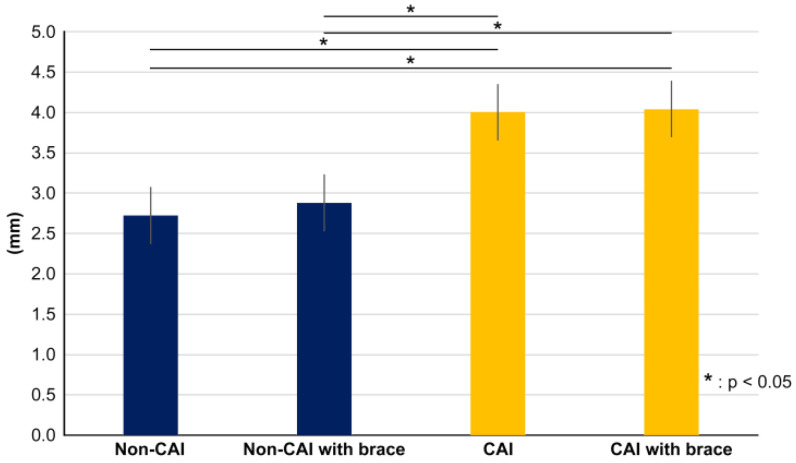
Graph showing the amount of change in medial–lateral direction of the center of pressure. The horizontal axis indicates each condition, and the vertical axis indicates the amount of change. There was a significant difference between the CAI and non-CAI sides, but there was no significant difference between with or without the brace.

**Figure 7 ijerph-20-05506-f007:**
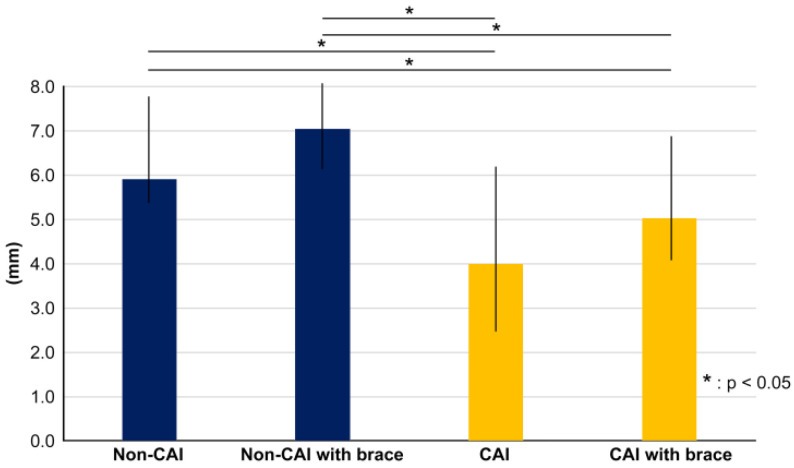
Graph showing the amount of change in anterior–posterior direction of the center of pressure. There was a significant difference between the CAI and non-CAI sides, but there was no significant difference between with or without the brace.

**Table 1 ijerph-20-05506-t001:** Ten athletes who have unilateral chronic ankle instability (CAI).

	CAI	Non-CAI	*p*-Value
N (foot)	10		-
Age (year)	20.5 ± 0.5		-
Height (cm)	180.1 ± 1.5		-
Body mass (kg)	83.0 ± 1.0		-
Cumberland Ankle Instability Tool	20.5 ± 0.5	28.0 ± 1.0	<0.05
Time since last ankle sprain (month)	3.0 ± 0.5	18.0 ± 2.5	<0.01

## Data Availability

The data supporting the study’s conclusions are accessible from the author.

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
