# Peer review of "Center of Pressure Deviation during Posture Transition in Athletes with Chronic Ankle Instability"

_ijerph, 2023, doi:10.3390/ijerph20085506_

Round 1

Reviewer 1 Report

Thank you for the opportunity to review your manuscript, “Center of pressure deviation during posture transition in athletes with chronic ankle instability”.

This is a clearly presented article. However, there are aspects that I consider should be improved.

1.     The study sample is very small (only 10 participants) and the non-inclusion of women is an important bias.

2. “In this study, all participants were male athletes who exercise regularly”. The type of exercise they do should be clarified.

3.     Therefore, it should be noted in the text that the small sample size and the fact that all subjects are men is an important limitation of the study. These results cannot be extrapolated to women because of the different anatomical characteristics of their lower limbs. Nor can they be extrapolated to non-athletes, due to their different capacity for postural control than patients who practice sport. These aspects should also be included in the conclusion of the study.

Figure 2: “Experimental procedures”. I think the figure would be more visual with a diferents color bar graph (for example). Also, the legend is not clearly legible.

Author Response

Dear Dr. Reviewer,

Thank you for your ongoing consideration of our manuscript for publication in IJERPH. We appreciate the time spent by you and the reviewers and believe the revised manuscript is improved. Below, we have addressed your and the reviewers’ comments.
We look forward to your editorial decision.

We thank the reviewer for the careful review of the manuscript.

1: The total number of subjects in this study was 186, of which 12.43% (23) were included in the CAI inclusion criteria. In addition, 43.2% (10 subjects) of them met the inclusion criteria for CAI in only one leg. This study recruited subjects with CAI in only one leg, which limited the number of subjects. However, we believe that the power, effect size, and significance level in the overall number of subjects are sufficient.
And also, women are not included as subjects because it has been shown that menstrual cycle modulates ankle ligament laxity. In order to eliminate different factors due to modulation of laxity, only male subjects were included in this study. Reference studies illustrating the above are as follows.
Yamazaki T et al.: A preliminary study exploring the change in ankle joint laxity and general joint laxity during the menstrual cycle in cis women. Journal of Foot and Ankle Research. 14:21, 2021

2: All male subjects in this study were athletes with Tegner Activity Score of Level 9 or higher. I have made an addition to the manuscript.

3: Thank you very much for your very important points.
The main purpose of this study is to evaluate the function of CAI, which is frequently observed in athletes. In addition, the subjects were selected to exclude structural factors of the ankle joint due to the menstrual cycle. However, all the points you pointed out are limitations of the study, and we have added them to the manuscript.

Figure 2: Thank you very much for pointing this out. We have corrected the image to a high quality image, so please check the image reciprocally.

Sincerely yours,

Reviewer 2 Report

Comments to the manuscript with ID ijerph-2291073 entitled: Center of pressure deviation during posture transition in athletes with chronic ankle instability. This is manuscript that analyze the Center of Pressure variation in patients with ankle instability performing flexion knee movements on a plantar pressure.

The manuscript is well written, but changes should be done. The text should be well justified

1.- Abstract:

Please add the sections in the abstract as the introduction, material and methods…

2.- Introduction:

Is well explain and explain the purpose and the hypothesis.

3- Participants and methods

Can authors add the reference or references about the goniometer reliability?

Was the goniometer measurement performed by the same researcher?

Can authors explained how was guided the arms of the goniometers?

How calculate the sample size? Please explain

Have all the subjects the knee alignments in sagittal and frontal planes??

How was the kind of the foot of the participants? Normal, flatfoot, pronated, cavus…..

Is the Footscan platform validated? Please add references

Point 2.3. Data analysis: please explain better the first paragraph. It does not understand.

Point 2.4. Statistics: please explain the statistical analysis. It doesn´t well explain.

4.- results.

Correct the tense “is” by “were”.

How calculate the p value in table 1?. Add abbreviations in table 1.

5.- Discussion

Can authors compare your results with other studies with genu valgus, genu varus and different kind of foot? Add references.

Author Response

Dear Dr. Reviewer,

Thank you for your ongoing consideration of our manuscript for publication in IJERPH. We appreciate the time spent by you and the reviewers and believe the revised manuscript is improved. Below, we have addressed your and the reviewers’ comments.
We look forward to your editorial decision.

We thank the reviewer for the careful review of the manuscript.

1: Thank you very much for pointing this out to me. However, upon reviewing the journal's submission rules, they do not require headings in the abstract. Therefore, we did not change the description.

2: As suggested, we have discussed this issue in the Introduction section. The following sentence was added at the end of the introduction「We hypothesized that the CAI group with the knee joint fixed in extension would have a significantly higher COP deviation than the Non-CAI group, revealing an unfixed invisible phenomenon.」

3: Measurements using the goniometer were taken by one measurer during the experiment. The goniometer promptly checked the angles of the knee joint and hip joint within 20 seconds of the subject standing on one leg. Before the measurement, the subject was given practice time to familiarize himself with the test maneuvers he was to perform. Failed attempts were excluded, and we believe that a certain level of reproducibility was ensured.
The total number of subjects in this study was 186, of which 12.43% (23) were included in the CAI inclusion criteria. In addition, 43.2% (10 subjects) of them met the inclusion criteria for CAI in only one leg. This study recruited subjects with CAI in only one leg, which limited the number of subjects. However, we believe that the power, effect size, and significance level in the overall number of subjects are sufficient.
Thank you very much for pointing out a very important point. We align the feet on the Footscan at the time of measurement for all subjects. From that perspective, we also believe that the knee joint is aligned with the knee in the sagittal and frontal plane.
No validation was performed on the foot morphology of the measurers. This study evaluates ankle joint function in athletes with CAI in only one foot. Because the number of subjects is not large, detailed classification down to foot morphology is not possible. However, we recognize that the results include the influence of foot morphology. We would like to add the influence of this factor to the study limitations.
Thank you very much for your valuable feedback, we have added a previous study on the accuracy and reliability of Footscan.

Data analysis: We would like to thank you sincerely for bringing up a very important point. It was a sentence that was difficult to understand, so I corrected the first sentence. We would appreciate your understanding.
「In this study, we tried to verify the COP deviation and examine the area and movement acceleration.」

Statistics: We appreciate the reviewer’s suggestions. We added sentences  because there was a lack of explanation. We would appreciate your understanding.

Result: Thank you very much for your important feedback.
We would appreciate it if you could check it as it has been corrected.

Discussion: No validation has been performed on the morphology of the knee joint of the participants. Also, previous studies have not verified the effect of knee joint alignment on COP. This is because this study excluded the influence of the knee joint and examined the influence of the ankle joint alone on COP. However, we recognize that the results do include the influence of knee joint morphology. We would like to add the influence of this factor to the study limitations.

Sincerely yours,

Reviewer 3 Report

strength: A study on the effect of knee braces on the COP of patients with CAI. In CAI patients, the application of a knee brace had a good effect on ankle stability, resulting in the improved center of gravity movement. More research is needed in the future.

weakness: This study is a single-group study. A further randomized study is needed.

1. LAS and CAI continue to be used interchangeably in the introduction. I think you can explain LAS only in the first paragraph. Ultimately, the research subject is CAI.

2. It would be better if you explain in detail the relationship between CAI and postural control ability in the second paragraph of the introduction.

3. Knee-joint function and hidden hip joint motion are presented for research purposes. In the introductory part, it is necessary to explain CAI and its relationship.

4. Please fill out the reliability and validity of the evaluation tool.

5. Is there no normality test?

6. Can you do a sample size calculation?

7. Please provide the name, country of manufacture, and manufacturer of the knee brace.

8. Results may be better due to the carryover effect. Please suggest ways to eliminate the carryover effect in the Procedure.

9. Please describe in more detail how knee braces affect COP in your review.

Author Response

Dear Dr. Reviewer,

Thank you for your ongoing consideration of our manuscript for publication in IJERPH. We appreciate the time spent by you and the reviewers and believe the revised manuscript is improved. Below, we have addressed your and the reviewers’ comments.
We look forward to your editorial decision.

We thank the reviewer for the careful review of the manuscript.

1: We appreciate the reviewer’s suggestions. Previous studies have also described the development of CAI due to repeated Lateral ankle sprains. We have included LAS as the pathogenic mechanism of CAI, which is the subject of this study. We would appreciate your understanding.

2: As suggested, we have changed the sentences. We would appreciate it if you could check the sentences.

3: Thank you for pointing this out. In the introduction, We showed why the knee joint function should be restricted in order to understand the pathology of CAI. we would appreciate it if you could confirm it.

4: Thank you very much for your valuable feedback, we have added a previous study on the accuracy and reliability of Footscan.

5: We would like to thank you sincerely for bringing up a very important point. We have added a paragraph on normality to the statistics. We would appreciate your understanding.

6: We thank the reviewer for the careful review of the point. The total number of subjects in this study was 186, of which 12.43% (23) were included in the CAI inclusion criteria. In addition, 43.2% (10 subjects) of them met the inclusion criteria for CAI in only one leg. This study recruited subjects with CAI in only one leg, which limited the number of subjects. However, we believe that the power, effect size, and significance level in the overall number of subjects are sufficient.

7: Thank you very much for pointing this out to us. We have added the details of the orthotic devices, as they were not described in detail.

8: We would like to thank you very much for your very important remarks. The estimated carryover effect in all statistical results is higher than the p-value. Therefore, the carryover effect was not statistically significant.

9: We thank the reviewer for the excellent comments. We added the point about knee brace in the conclusion area. 

Sincerely yours,

Round 2

Reviewer 2 Report

Authors have done all the changes suggested in proper mannner ending a great article.

Congratulations.